# Architectural Heritage Preservation for Rural Revitalization: Typical Case of Traditional Village Retrofitting in China

**Kai Xie, Yin Zhang * and Wenyang Han**

School of Architecture, Southwest Minzu University, Chengdu 610225, China

* Correspondence: cdzhangyin@163.com; Tel.: +86-134-8891-8589

**Abstract:** With the massive urbanization and industrialization in China, the paradox between fast modernization and cultural preservation arouses challenges on new insight into green construction and sustainable development strategies throughout the nation. Particularly, how to strike a balanced cultural–modern rural revitalization has become a research priority, especially for cultural and historical villages in relatively under developed western regions. In this paper, taking Gaotunzi Village, a traditional ethnic village in western high-altitude plateau as an illustrative example, the typical green design manner and optimization strategy are proposed for cultural and architectural heritage preservation. The detailed architecture and structure design is conducted for both old temple retrofitting and new museum construction, with consideration for retaining traditional building colors, styles, and materials. Moreover, land use planning is demonstrated for local public space and services upgrading. The overall design strategy has been taken into practice for the local renovation construction program. The preliminary results indicate that this green retrofitting idea and approach are quite locally feasible for sustainable rural revitalization with local heritage conservation, including traditional wooden buildings, Buddhism belief, Tibetan icons, and ethnic symbolic culture preservation and promotion. This work can provide a typical design reference and application prototype for rural construction and modernization with local heritage preservation considerations, especially for those traditional villages in developing countries.

**Keywords:** architectural heritage; rural revitalization; traditional village; building design; culture conservation

## 1. Introduction

With the massive urbanization and industrialization in China, the paradox between fast modernization and cultural preservation arouses challenges on new insight into green construction and sustainable development strategies nationwide [1]. The rapid social–economic development inevitably creates a growing disparity between the lifestyles of the residents of traditional villages and the modern lifestyles [2]. In order to improve the residential living standard in traditional villages while effectively protecting the local traditional architectural heritage and culture, it is necessary to efficiently combine the local folk customs and religious cultures, and to take the strategy for the revitalization of the countryside in China as the guiding ideology, so as to organically combine the suitable renovation of the traditional villages with the process of urbanization.

The VerSus project, conducted in Europe, puts forth the methodology for the identification and analysis of the vernacular heritage based on three main principles of conception: environmental, sociocultural, and socioeconomic. General objectives and needs related to each sustainable scope were established, together with strategies learned from vernacular heritage projects, which are of great value [3–5]. An international charter for the conservation and restoration of monuments and sites proposes that the purpose of preservation is not only to preserve a historical relic to satisfy people's nostalgia for historical culture, but also to continue our culture and life from a material perspective, and the use of historical

sites is always beneficial to their preservation [6]. It is further pointed out that efforts to conserve, restore, and reuse existing historic areas and architectural monuments be integrated with the process of urban development in order to assure their proper financial support and continued viability [7].

Development and renewal are irreversible, and modern technology and life have permeated every corner of the world. Replacement, reconstruction, and environmental renewal of architectural heritages that no longer meet the needs of contemporary society can enable the architectural heritage to re-reflect its intrinsic value and achieve new identity, which is a necessity for society [8]. In the 1980s, modern building materials and structures such as glass and metal began to be widely used in the reuse, creation, and new construction of architectural heritage, reflecting the aesthetic characteristics and technological charm of modern architecture. The denotation and connotation of architectural heritage protection have also been expanded, from protecting individual buildings to protecting historical architectural complexes, and from protecting tangible material heritage to protecting intangible cultural information. ICOMOS states that our task is to protect all the architectural heritage of humanity, not just world heritage [9].

It has been becoming increasingly important to protect, revitalize, and reuse traditional architecture and cultural heritage by applying the cultural advantages of traditional ethnic minority villages themselves [10]. Combining the modern lifestyle with the architectural renovation of traditional villages not only plays a protective role but also arouses new vitality in traditional villages, promotes the development of tourism, and thus promotes the sustainable development of traditional villages, which is of great significance in promoting the internal economic cycle of the country [11]. Establishing a comprehensive and systematic ethnic, cultural, and society-involved concept of protection has become a contemporary consensus [12]. The environment in which architectural heritage exists has changed, as have the living structures and lifestyles within the buildings. The most appropriate way to protect them is to revitalize old buildings and old neighborhoods in a new environment, rather than turning them into "specimens" [6]. The combination of modern technology and lifestyle with the transformation of traditional villages can not only effectively protect the architectural structures but also stimulate the vitality of traditional villages, improve villagers' living standards, and form sustainable development and internal economic cycles within the country [13]. China is a multiethnic country with a long history; different ethnic groups have their own special cultural symbols, cultural heritage, and ancient buildings to be preserved, leading different regions to have unique characteristics, providing a rich research blueprint for the protection of architectural heritage and thus further promoting the revitalization of villages [14].

Over the past few years, there has been a significant increase in studies on the topics of architectural heritage conservation, sustainable land use planning, construction renovation, city regeneration, and rural revitalization [15–17]. Kontokosta et al. [18] considered that the protection of traditional minority architecture and culture is a very important concern in the process of urban development, and emphasize the organic integration of traditional minority culture into modern cities, so as to make it show new cultural vitality. Young [19] studied the problems facing the protection of architectural heritage of traditional villages in the context of rural revitalization, and put forward realistic paths of protection and development suggestions. Mohd-Noor [20] and Lee et al. [21] pointed out the dilemma of heritage protection and considered the social-level influence. Rahaman [22] summarized the current situation of domestic vernacular architectural heritage protection and the direction of future research in the light of the progress of foreign research on architectural heritage protection. Kee et al. [23] studied the relationship between adaptive reuse of heritage and the authenticity of cultural heritage with social and economic benefits. Then, Rong et al. [24] pointed out that the restoration and reuse of architectural heritage can effectively preserve vernacular components and cultural narratives in the context of the sustainable development strategy of traditional architecture. Nguyen et al. [25] indicated that architectural heritage can create core economic dynamics of the heritage site while

shaping the subjectivity of the community, and the conservation of architectural heritage and the economic dynamics of the conservation site were studied in conjunction with the direction of the economic dynamics of the conservation site.

Furthermore, with emphasis on architectural heritage conservation and construction development, a variety of scholars have focused on examining regional and local rural development [26,27]. Wang et al. [28] revealed that the expression of intangible cultural elements in residential architectural heritage was insufficient among the current world heritage sites, and put forward architectural landscape protection and inheritance coping strategies based on the principle of authenticity. Yu et al. [29] studied the relationship between the spatial layout and architectural structure of the characteristic dwellings in the Tibetan area and the historical conditions, humanistic heritage, and geography and climate, reflecting the adaptability of the terroir. Zhang et al. [30] explored the architectural space of a Tibetan village and proposed that the protection of traditional architectural heritage requires "integration and transformation".

Moreover, Allam et al. [31] reported that the renovative and regional adaptive design should not only respect and protect the traditional architecture, but also fully consider possible reusage and recycling. Kuzior et al. [32] studied the spatial characteristics of traditional villages in terms of landscape resources. Rangarajan et al. [33] examined the potential relationship between traditional villages and pre-religion in remote areas. In addition, Duca et al. [34] stated the high necessity of public service and outdoor space optimization design in living standard improvement for most city or rural regeneration programs.

Although existing research has provided many methods and ideas for the field of green sustainable architecture and planning design, it has mainly focused on the direction of new construction in developed cities and regions, such as Copenhill in Denmark, Gurdau Winery in Czechoslovakia, and The Farmhouse in Austria. Therefore, we need to consider how to achieve a balance between modernization and cultural preservation in traditional rural areas with regional and ethnic cultures, under the increasing pressure of urbanization. What are the adaptive design and planning standards for local rural revitalization in the field of land use and construction? In practical engineering applications, how can modern architectural technologies with cultural symbolic significance be integrated into architectural heritage preservation and renovation plans?

To tentatively provide the answering mode or prototype to the aforementioned academic questions, this paper takes the temple retrofitting project of a traditional ethnic village, located in Songpan county, Western China as a typical illustrative example. The main works are structured in the following sections. (1) Field investigation on current status of local land use and building construction level is conducted to reveal the detailed problems, with emphasis on the demands of architectural heritage preservation and the analysis of regional cultural symbols. (2) Detailed architecture and structure design is demonstrated from three aspects: old temple retrofitting, new museum construction, and upgrading outdoor public space and services. (3) Furthermore, the significance of traditional–modern integration is discussed in building design and space planning strategy with local cultural conservation considerations. This work can provide a typical design reference and application prototype for rural construction and modernization with local heritage preservation considerations, especially for those traditional villages in developing worlds.

## 2. Methods

### 2.1. Illustrative Village Overview

Songpan, a county in northwestern Sichuan province, is located on the eastern edge of the Qinghai–Tibetan Plateau, and in northeastern Aba Tibetan and Qiang Autonomous Prefecture (Figure 1). More than ten ethnic groups reside here, with Tibetan, Qiang, Hui, and Han being the main ones. Gaotunzi Village in Songpan County is located in northeastern Shili Hui Ethnic Township, 5 km from the ancient city of Songpan and 7.5 km from Chuanzhusi Town. It is adjacent to the "Jiuzhai Huanglong" World Natural Heritage

Site and Min River Source National Wetland Park, and is the only route to the National Long March Cultural Park Monument.

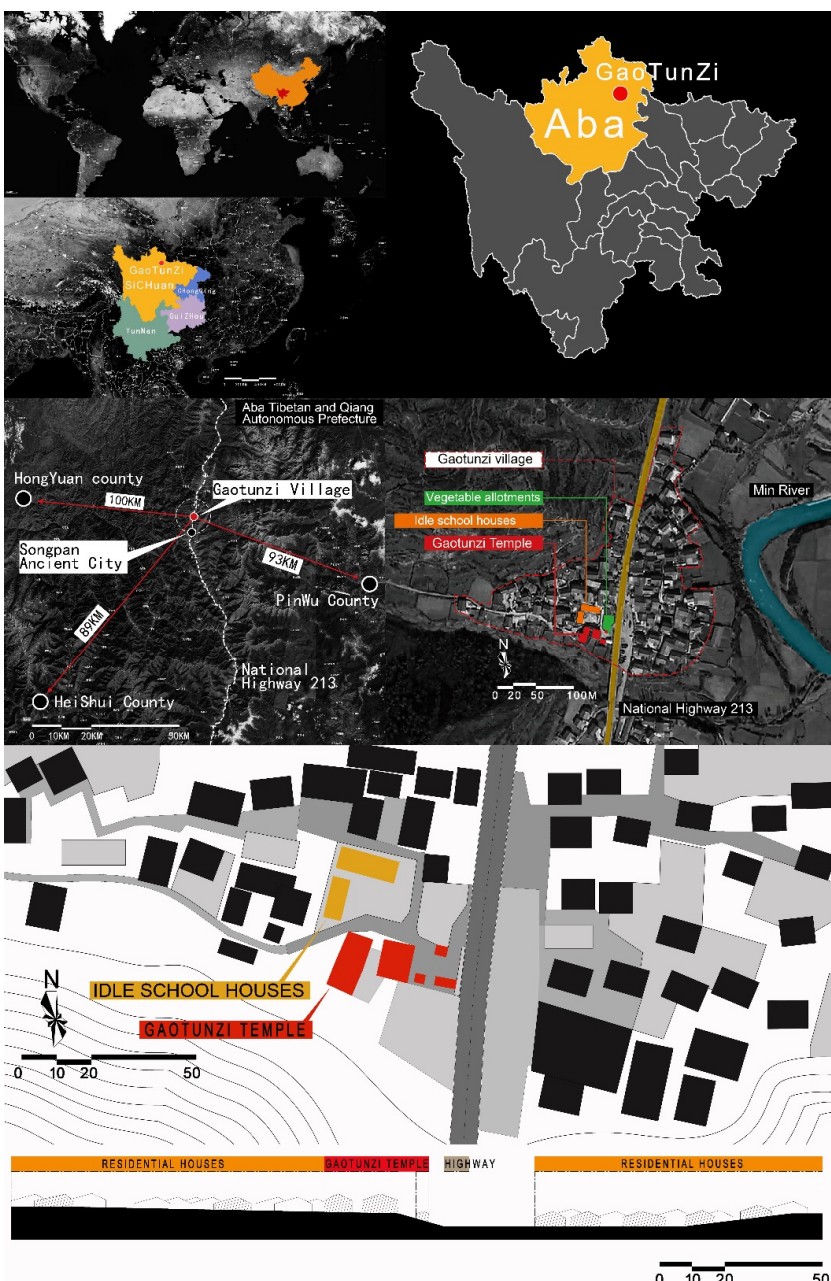

**Figure 1.** Location map of Gaotunzi Village in Sichuan, China (drawn by the authors).

Gaotunzi Village is located at the junction of the Amdo and Jiarong Tibetan areas, serving as the gateway to the northwestern Sichuan and Gannan Tibetan regions. The core area of the village covers approximately 12.63 hectares, with 151 households and a population of 625, consisting of a multiethnic community with the Jiarong Tibetan people as the main group. The Jiarong Tibetans, who speak the Jiarong language as their mother tongue, are engaged in agricultural production and are referred to as "Rongba" (agricultural people) in Tibetan [35]. The village is situated in the Min River Valley, surrounded by mountains on the east and west, with 1000 mu (Chinese measurement that is commonly 0.165 acre or 666.5 square meters) of arable land, 5405 mu of grassland, and 800 mu of forest, creating a landscape of intertwining pastoral and riverine areas, typical of the flat-type Tibetan natural villages on the western Sichuan Plateau.

The village's terrain is characterized by a lower middle section surrounded by gradually rising land, with a height difference of 6 m between the highest point on the northern edge and the lowest point (Figure 1). The overall form of the village is a clustered structure radiating outward from Gaotunzi Temple at its core. National Highway 213 divides the village into eastern and western parts, with Gaotunzi Temple located in the western half, adjacent to the national highway [36].

During Tang Dynasty (618–906 A.D.), the court stationed troops and smelted in Songzhou (former name), forming the prototype of Gaotunzi Village. For thousands of years thereafter, Gaotunzi Village has been a military stronghold in western Sichuan and a trade channel for tea and horse exchanges. Xue Tao, one of the four great female poets of the Tang Dynasty, was exiled here. According to records, Xue Tao wrote over 80 poems here, and many legends of Xue Tao, such as the "Pure Immortal Flower", are still circulating in the countryside. The villagers have diverse and concurrent beliefs, and Taoism and Tibetan Buddhism are enshrined simultaneously in Gaotunzi Temple. The Han culture from the soldiers, the legend of Xue Tao, the culture of poetry, ethnic folk customs, commercial culture, and religious culture have accumulated here, forming the historical context and ethnic beliefs in a collective memory of the villagers, and leaving behind cultural relics such as Gaotunzi Temple.

### 2.2. Field Research on Current Situation

Although Gaotunzi Village has a long history, due to severe lack of data, detailed information and settlement morphology of Gaotunzi Village in history cannot be verified. All the buildings in the current Gaotunzi Village were constructed during the past thirty years, with modern brick and concrete houses, and there are no historical architectural relics remaining in the village. In the past decade, villagers have randomly used modern building materials such as cement, red bricks, and colored steel tiles in the construction of houses without any overall planning, causing a drastic change in the village's appearance. The former kindergarten and primary school buildings in the village are abandoned and idle, and the only public space for villagers' activities is a makeshift pavilion in front of Gaotunzi Temple, which also serves as the village committee's information board. There is a lack of public service buildings and spaces for villagers' activities, and there is no parking lot in the village, leading to arbitrary parking of vehicles, which seriously affects the village's appearance.

Based on oral history from local residents, the multiethnic population, mainly consisting of the Jiarong Tibetan people, has lived around Gaotunzi Temple for generations. Gaotunzi Temple has always been the core of the village, serving as the most important space for the villagers' religious activities, daily life, festival celebrations, neighborhood interactions, and ethnic integration. According to the oral recollections of local elders, Gaotunzi Temple was first built in the Guangxu Period (1875–1908) of the Qing Dynasty and has undergone multiple reconstructions. Due to insufficient awareness of its cultural value and a lack of preservation efforts, the original layout of the temple has been completely lost. The front hall of the temple was built in the 1970s, and the buildings in the rear courtyard were constructed within the past decade. As a result, there is no way to verify the architectural style, materials, and colors of the original Gaotunzi Temple before its reconstruction.

As stated in "Manual of Living Heritage Conservation Methods" by the International Research Center for the Protection and Restoration of Cultural Heritage in 2009, heritage should be regarded as a "cultural process". Gaotunzi Temple has always been a center for villagers both culturally and religiously. It has been a historical witness and has been a prominent part of collective memory for the villagers. Despite its relatively newness, thus not qualifying as a historical landmark, it has undoubtedly been involved in a broader process of ethic culture and historical changes. In this sense, Gaotunzi Temple should be valued and protected as part of the culture heritage.

The space inside Gaotunzi Temple main hall is limited, and some Buddha statues and Earth God shrines are set up in simple sheds built on both sides of the lane in front of the main hall. On the north side of the backyard is a makeshift Buddha hall, while on the west side is a double-slope simple residential building serving as the monks' living area. The living and worshipping spaces overlap, with debris scattered around in a dilapidated environment. Villagers and pilgrims come to pray and worship, while the elderly in the village chat and relax inside the simple village committee information bulletin booth in front of the hall. The different functions of the front space intersect due to informality and temporality, which leads to a mixed and chaotic worshipping place and is very hard to manage (Figure 2).

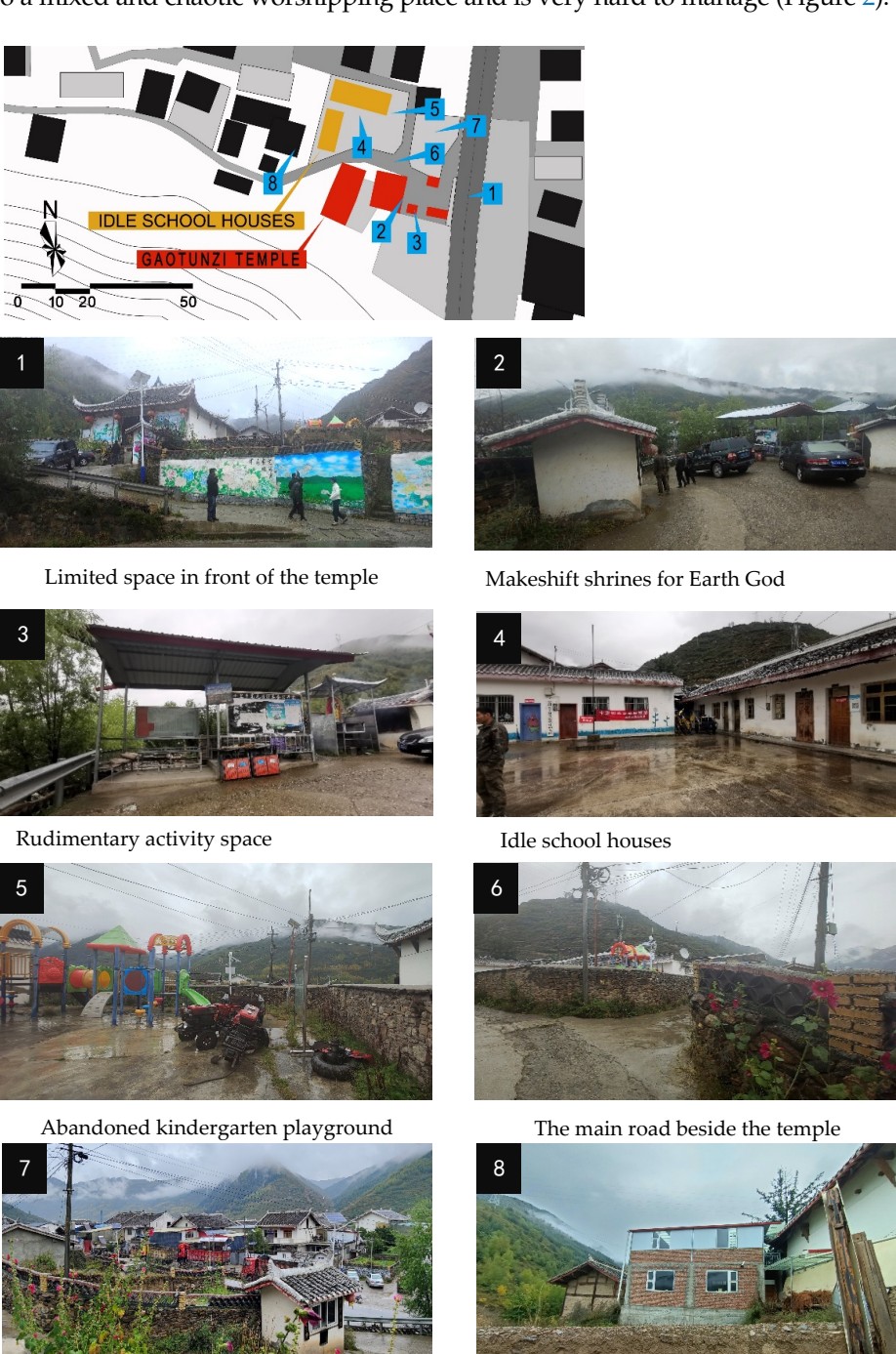

**Figure 2.** Practical images of current underdeveloped construction status in Gaotunzi village (pictures taken by the authors).

　　　In 2022, the design team, village committee, and villagers engaged in discussions at the request of the village committee. The community and villagers expressed their desire to renovate Gaotunzi Temple and repurpose the abandoned buildings and temporary vegetable allotments in the village to enhance the village's environmental quality and provide more public spaces for the residents. Considering the fact that female poet Xue Tao from Tang dynasty was once exiled here, the Gaotunzi village community hopes to design a dedicated architectural space to showcase her poetry and legends that circulate in the village. The design scheme described in this article is a design result submitted to the village government, and the project is currently in the preliminary implementation stage.

*2.3. Local Cultural Symbolic Icon Analysis*

　　　The mandala, Mount Meru, and Pure Land patterns are common themes in traditional Tibetan patterns. The Vajradhatu (Diamond Realm) mandala is characterized by a converging cross-shaped axis, while the Garbhadhatu (Womb) mandala is characterized by a layout of concentric squares, both featuring a pyramid-shaped spatial prototype and implicit cross-axis characteristics. The Mount Meru pattern presents multiple concentric circles, with a central towering structure gradually decreasing in height towards the periphery. These patterns represent the imagination and aspirations of the Tibetan ancestors for the cosmic world and the ideal living environment, serving as "prototypes" of architectural space that are constructed through shared visual perception. This pattern concept deeply influences the spatial layout of Tibetan settlements, with the temple typically serving as a visual reference point at the center, emphasizing a sense of stability in the central composition, with residential buildings radiating and expanding around the core space.

　　　Religious facilities such as temples, lama towers, and prayer rooms are constructed in the most advantageous locations within the settlement, and they are generally not adjacent to water for fear of flooding. The spatial positioning of other material settlement elements with different attributes is relatively flexible. Settlement areas often have clear boundaries or restrictions. The natural landscape forms the outer foundational environment of the settlement, while cultivated land, grasslands, mountains, water bodies, and buildings are the basic elements that outline the boundaries of the settlement.

　　　Residential buildings are basic components of a settlement, and due to requirements for privacy and security, they normally do not intersect with other functional components of the settlement. The architectural layout of traditional villages typically revolves around a central square, with houses arranged in concentric circles around it. The central square serves as a place for community activities and gatherings, symbolizing the cohesion of the community. The houses are arranged based on family and kinship relationships, creating a closely connected community. The material elements of the settlement serve different functions based on the activities of the residents. Elements such as parking lots, recreational spaces, lama towers, prayer rooms, temples, and other public spaces are clustered around the central buildings, forming a circular layout. These spaces are also important places for communication and interaction among community residents.

　　　The traditional residential architecture of the Jiarong Tibetan people is an important cultural heritage in the southwestern region of China, with unique architectural features and cultural significance. Jiarong Tibetan villages are often built on sunny and wind-sheltered slopes, with several to dozens of households forming a village. The residential buildings of the Jiarong Tibetans usually have different functional areas, such as living area, kitchen area, ritual area, and storage area, which are divided into separate zones. The architectural layout typically revolves around a central courtyard, with various functional areas surrounding it. Each area has its specific design and layout to meet the needs of family life. The courtyard serves as a place for family activities and social interactions, as well as a symbol of family cohesion. The houses are arranged based on family and kinship relationships, thus forming a close-knit community (Figure 3).

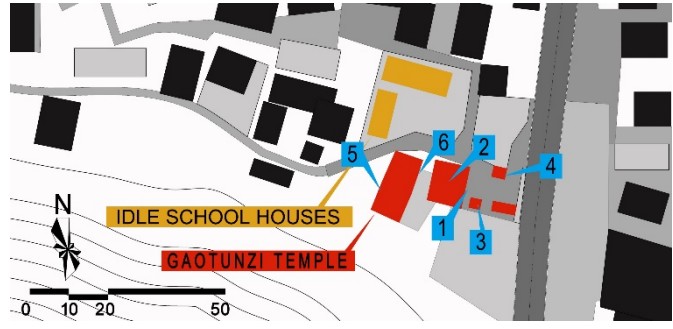

| | |
|---|---|
| 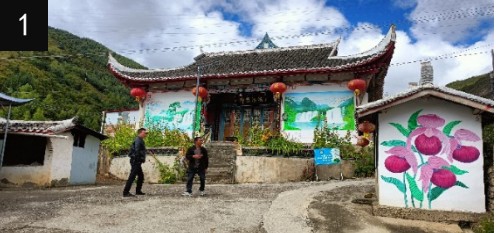 Main Buddha hall in the temple | 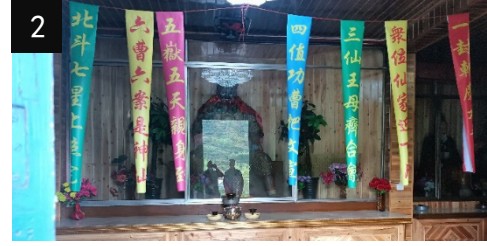 Interior of the main hall |
| 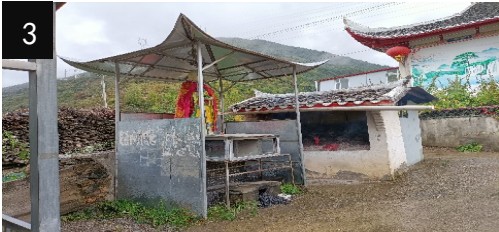 Makeshift incense-burning stall | 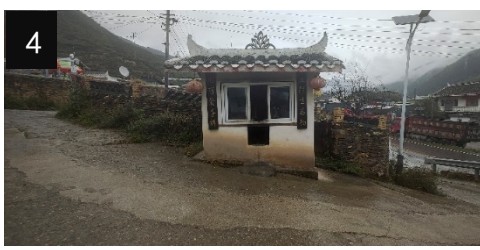 Temporarily built Earth God shrine |
| 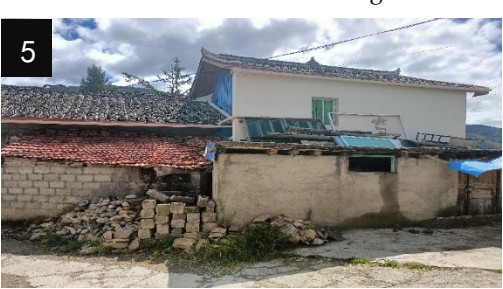 Monks' living area (partly collapsed) | 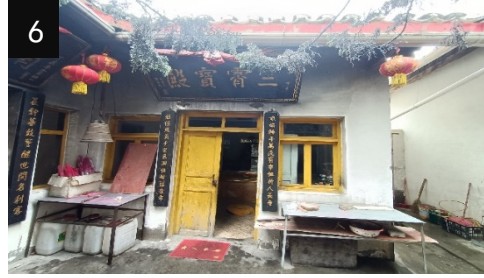 Open-air kitchen beside the main hall |

**Figure 3.** Practical images of local cultural and living styles in Gaotunzi village (pictures taken by the authors).

The construction of Jiarong residential houses utilizes local materials, with yellow clay used as the adhesive to build stone walls. The stones are stacked like bricks, with stones of varying sizes arranged to form the walls. Large stones are placed in a circle, leveled with small stones and clay, and then another circle is built on top, layer by layer, resulting in walls that are wider at the base and narrower at the top, with a slightly trapezoidal shape. This construction technique of rammed earth or stone walls provides good insulation and thermal insulation, adapting to the climate characteristics of the high-altitude region. The roofs of the buildings are supported by large timber beams and covered with a layer of mixed wood and soil. Stone slabs or wooden tiles are used to cover the roof, making it suitable for seismic and climatic conditions in the area. Residential buildings are often decorated with painted murals and carvings. Murals depict mythical legends, plants and animals, and abstract patterns, with vibrant colors. Carvings are commonly found on doors, windows, beams, columns, and furniture, showcasing their craftsmanship and artistic expression.

The Songpan region is home to many Diaolou buildings, with some famous ones still standing today, such as the 85 Diaolou Buildings in Danba Tibetan Village, groups of Diaolou Buildings in Jiaju Tibetan Village, and the twin Diaolou Buildings in Jiagenba Village, Xinduqiao Town. The construction of Diaolou buildings is closely associated with warfare. In ancient times, Songpan frequently experienced wars, and the Tibetan and Qiang ancestors used stones and stone slabs to build three- to four-story houses. Due to their resemblance to fortresses, they were named Diaolou. Diaolou served as "military bases" to resist enemies, allowing occupants to overlook and defend against the enemy from a higher vantage point. Diaolou can be standalone structures or combined with residential buildings, playing a leading role in the planning of villages and enriching the skyline of settlements. Gaotunzi Temple is a witness to historical changes and the integration of Han Chinese and Tibetan culture, and the specific form of the building back then cannot be verified.

The current Gaotunzi Temple leans against Gaotunzi Gully and faces the Min River, with a Han Chinese-style single-eave resting hill roof. The main hall has a width and depth of three bays, with a double-groove plan. The front hall has an open door in the middle, while the solid walls on both sides feature painted murals in Han Chinese and Tibetan style. The temple houses various religious deities from Buddhism, Taoism, and other religions. The interior space of the hall is cramped, with some Buddha statues and Earth God placed in simple shrines on both sides of the front driveway. In the backyard, there is a simple Buddhist hall on the north side, a subsidiary double-sloped residential building for the monks on the west side, and an open-air kitchen on the east side. The living and worship spaces overlap, with items scattered around in a deteriorated environment.

*2.4. Retrofitting Strategy and Design Core*

With the aim of rural revitalization in Gaotunzi Village, the design adopts an "acupuncture-esque" renewal strategy, which is more reasonable compared to a comprehensive demolition and reconstruction plan. Although the architecture of Gaotunzi Temple follows a Han Chinese style, it has always been the spatial core of the village layout. The design focuses on the protection of Gaotunzi Temple and the renewal of the temple space, and based on this, it aims to transform and enhance the core space of the village, expand and upgrade public spaces for the villagers, and provide room for relaxation. Similar to acupuncture in traditional Chinese medicine, by enhancing key nodes, it aims to lead and gradually drive the overall transformation and development of the entire village.

From October 2022 to April 2023, the design team visited Gaotunzi Village multiple times and stayed for 10 days altogether. The village layout is constructed based on the onsite survey and satellite images. Data such as the village elevations, site location measurements, and samples of colors and architectural materials were collected. The architectural techniques of the Jiarong Tibetan people are mostly recorded in the literature, such as Songpan County History.

As Figure 4 shows, the redesign and planning of the village respect the commanding position of Gaotunzi Temple within the settlement. Taking into account the community's needs and renovation suggestions, the design utilizes the idle school land within the village to transform and design it into the Xue Tao Cultural Museum and Village History Museum. It integrates the temporary vegetable allotments on the southeast side to establish an activity center for the villagers. These buildings incorporate elements of faith, history, and leisure to form the core of the settlement.

The design extracts the spatial patterns of traditional Tibetan mandala and Mount Meru patterns, and based on terrain elevation differences and sightline analysis, a three-fold space comprising worshiping, living, production, and natural elements is formed: the core being the public space of Gaotunzi Temple, Xue Tao Cultural Museum, and Village History Museum; the second circle consisting of residential and community activity spaces; the third circle encompassing production spaces such as farmland, fields, and forest areas on the outskirts of the settlement. The peaks of the mountains behind Gaotunzi Gully

on the north, the courtyards of temples, and the mountain range on the south form a clear north–south spatial axis. Through the analysis of terrain height differences and sightline relationships, the Diaolou building and temple become the high points of the settlement, serving as focal points for converging sightlines and social activities. This creates a heterogeneous and cohesive spatial character within the homogeneous settlement, consolidating and strengthening the advantage of the core public space in vertical space and skyline design. The layout pattern of the settlement's spatial arrangement forms a coupled relationship with the traditionally concentric and hierarchically distinct patterns.

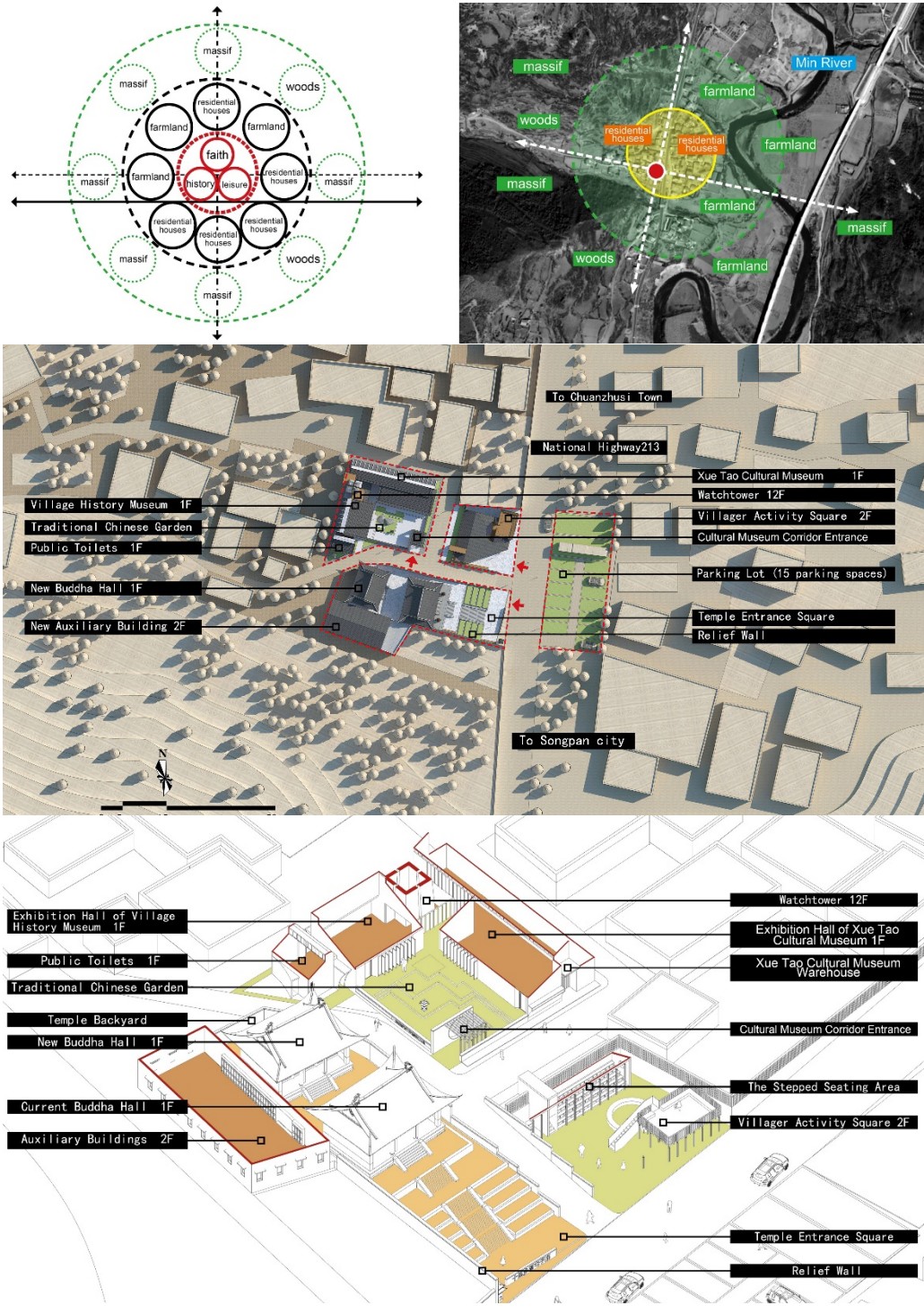

**Figure 4.** Overall land use planning and functional analysis of the Gaotunzi Village renovation.

The plan proposes to restore and protect Gaotunzi Temple and its associated buildings in an axial symmetric manner that is typical of traditional temples. It also suggests the construction of the Xue Tao Cultural Museum, Village History Museum, and Villager Activity Center, integrating small-scale pitched roofs into the village texture to preserve the regional Jiarong Tibetan architectural style. Additionally, it recommends adding a watchtower as a landmark building to highlight Tibetan cultural presence. The design proposes that the existing residential buildings in the village maintain their original double-pitched roof form, with a slope controlled between 10–15 degrees (measured onsite), while a few buildings may adopt a flat or combination of flat and pitched roof forms to create a unified texture and color scheme. The roofs of ordinary buildings should be covered with dark gray tiles. Key buildings, such as Gaotunzi Temple, should retain their special architectural form, such as the through-ridge gable roof. For Tibetan residential buildings, the angle of the first-floor walls should be controlled at around 5 degrees, with a narrower width at the top and a wider width at the bottom. Stone masonry should be used for the exterior walls to enhance seismic resistance and regional characteristics. Some exterior walls can be coated with textured paint, while materials such as marble, ceramic tiles, and antique bricks should be prohibited for the exterior walls.

## 3. Results: Detailed Optimization Design

### 3.1. Old Temple Renovation

In order to ensure authenticity and integrity of the main Buddha hall of Gaotunzi Temple, the design proposes to preserve the existing main hall building in its entirety. It also suggests the construction of a new, more modest Buddha hall in the second courtyard behind the main hall to house Buddha statues and Earth Gods that are currently displayed in makeshift sheds in front of the temple. Additionally, the existing residential buildings for monks on the west side of the backyard are expanded and refurnished into an auxiliary building, with the open-air kitchen on the east relocating into it. It will significantly improve the monks' life and enhance the temple's structural order between ritual and daily life.

The design utilizes the four-meter height difference from the temple entrance to the national road, transforming the steep slope of the village entrance ramp connecting to National Highway 213 into a ceremonial space in front of the temple (Figure 5). This corrects the steepness in the current turning ramp and establishes a horizontal connection between the longitudinal old street of the settlement and the national highway. The design removes the existing flower beds, makeshift pavilions, village committee information boards, and temporary incense-burning rooms in front of the main hall, relocating the statues and Earth God shrine to the newly built main hall. The design includes a shadow wall at the entrance of the front area in the main hall, and a sculptural wall is set on the west side of the entrance square.

A new wall is built along one side of the village road, enhancing the vertical perspective depth that highlights the spatial axis, while also blocking the neighboring residents' vegetable gardens, making the sacred space purer and more concise. The design incorporates terraces and flower beds at different levels to gradually divide the terrain height difference, restoring and strengthening the sense of axis in the traditional ceremonial space. It is not only intended to become a public space for the villagers and visitors, but also to reintegrate the architectural traces from different historical stages, highlighting the collective memory of Gaotunzi Temple in the lives of the villagers.

### 3.2. New Construction

As an important representative figure of Han poetry and culture, Xue Tao's story of exile and residence in Gaotunzi Village has been passed down through generations and has become a historical witness to the cultural exchange and integration between Han and Tibetan cultures in Gaotunzi Village and the Songpan area. The village authorities have requested the construction of a new exhibition space in the core area of the village to showcase Xue Tao's poetry and culture as well as the Jiarong Tibetan culture. The

L-shaped building on the north side of Gaotunzi Temple was originally a primary school, later converted into the village committee's office, and became a kindergarten in recent years. Due to a decrease in student enrollment, it is now vacant. This area is relatively high in altitude and is the most flat and spacious land within the settlement. Villagers often use it for drying agricultural products and for leisure and communication. Taking into account the spatial layout of the settlement, the design plans to transform this piece of land into the Xue Tao Cultural Museum and the Village Cultural Museum (Figure 6).

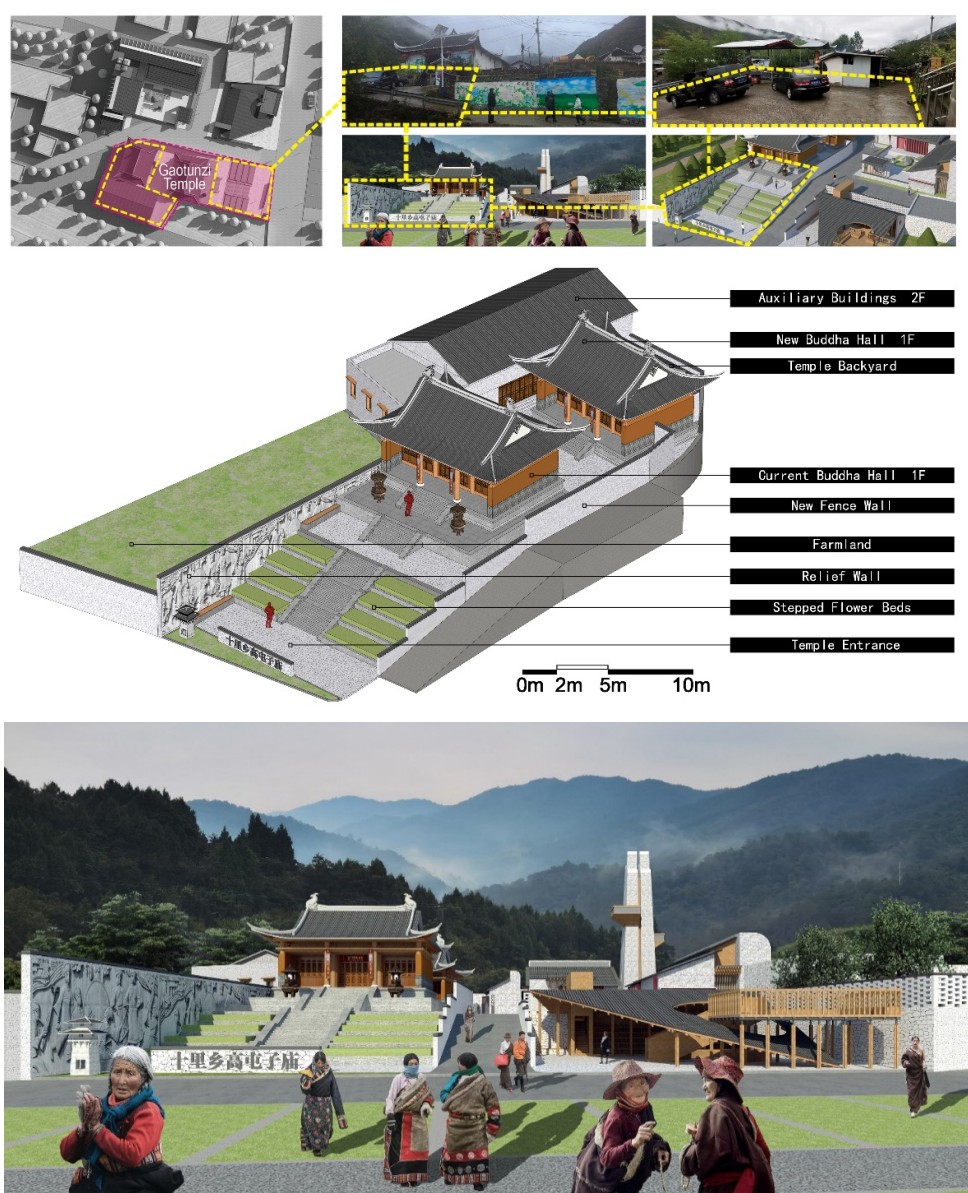

**Figure 5.** Gaotunzi Temple renovation design details (rendering by the authors).

With the Xue Tao Cultural Museum as the main building, the Village Cultural Museum, public toilets, watchtowers, and courtyard are enclosed to form an architectural courtyard, integrating into the settlement in a small-scale manner. The design attempts to blend Han and Tibetan architectural forms, combining ritual, poetic, humanistic elements in an open and coordinated way. The design adopts a courtyard layout, with the shadow wall, corridor, and Han-style landscape courtyard at the southern entrance reflecting Han architectural style. The low walls make the courtyard open and transparent, while the vertical pink lattice on the building facade symbolizes the light and romantic sentiments of Xue Tao, a female poet. The rough stone walls and sloping roofs of the building echo the "diaochao"

(fortress nest) form of stone and wood structures in Jiarong Tibetan dwellings. At the junction of the Xue Tao Cultural Museum and the Village Cultural Museum, a Tibetan-style viewing watchtower is designed, ensuring the visibility of the entire settlement and the national highway, highlighting the unique regional and outwardly symbolic characteristics of Sichuan–Tibetan architecture.

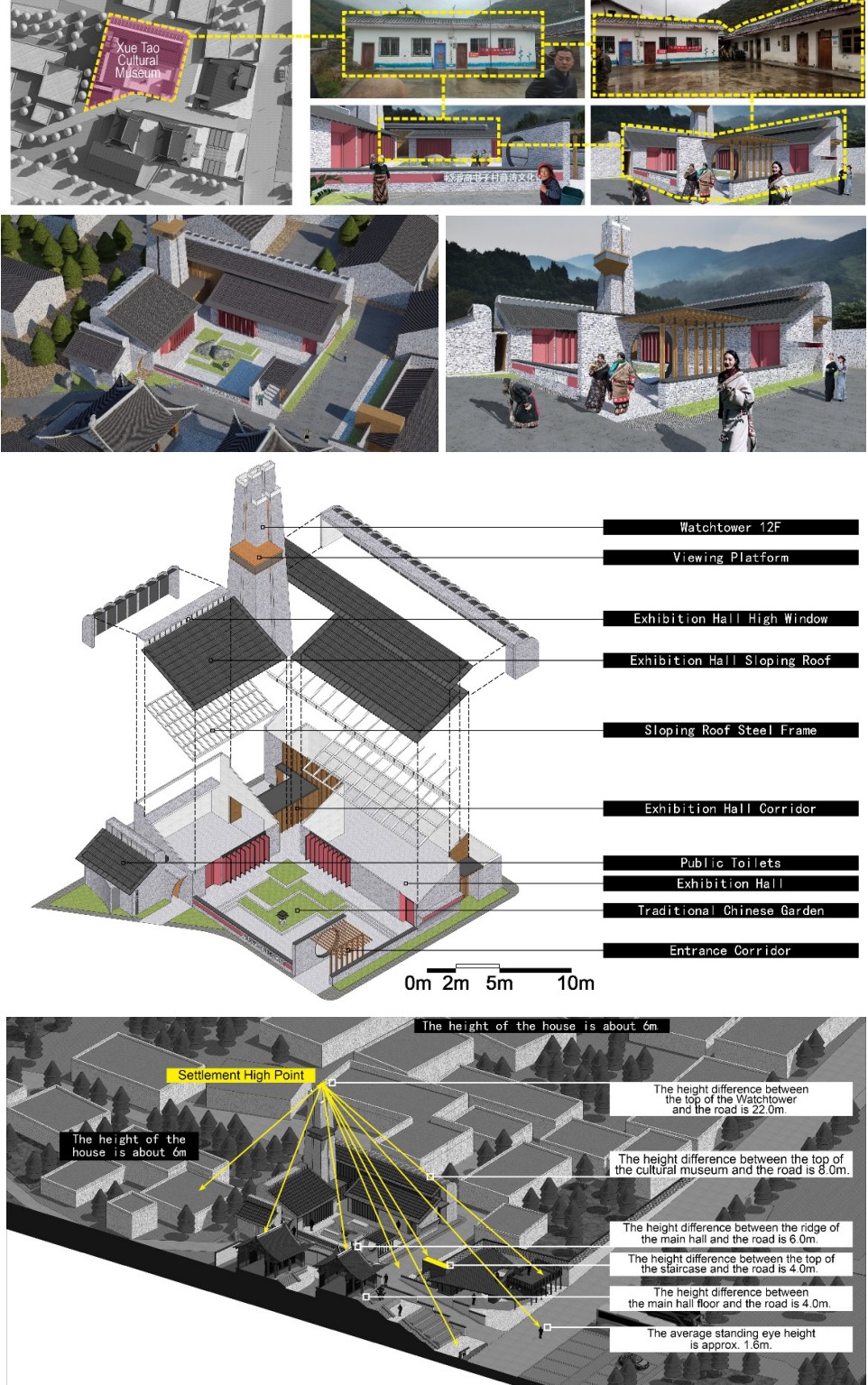

**Figure 6.** New construction of the Xue Tao Cultural Museum design details (rendering by the authors).

### 3.3. Public Space Reactivation

The only public space for villagers' activities is a simple shed of less than 4 square meters located at the information bulletin board in front of Gaotunzi Temple. The design chooses the farmland on the east side of the kindergarten near the national road to create a rural cultural activity center and provide a space for public communication, relaxation, and leisure for the villagers. By demolishing the unused area on the east side of the kindergarten and the retaining wall, the height difference of the ground is leveled with National Highway 213, creating a design that opens the visual corridor from inside, attracting the attention of passing tourists on the national highway with a vista of leisurely Tibetan village life.

Taking advantage of the 4 m height difference in the terrain, a sunken platform is created, covered by small blue-tiled roofs, forming a large unobtrusive gray space. Under the roof structure, wooden stepped seating is designed for villagers to rest and chat. The stepped seating can also serve as a podium for village meetings, and the barriers of the seating can be used to display and sell specialty agricultural products, promoting sales activities. This space serves as a place for casual conversations among villagers and also as a commercial space for villagers to sell their specialty agricultural products from Gaotunzi Village (Figure 7).

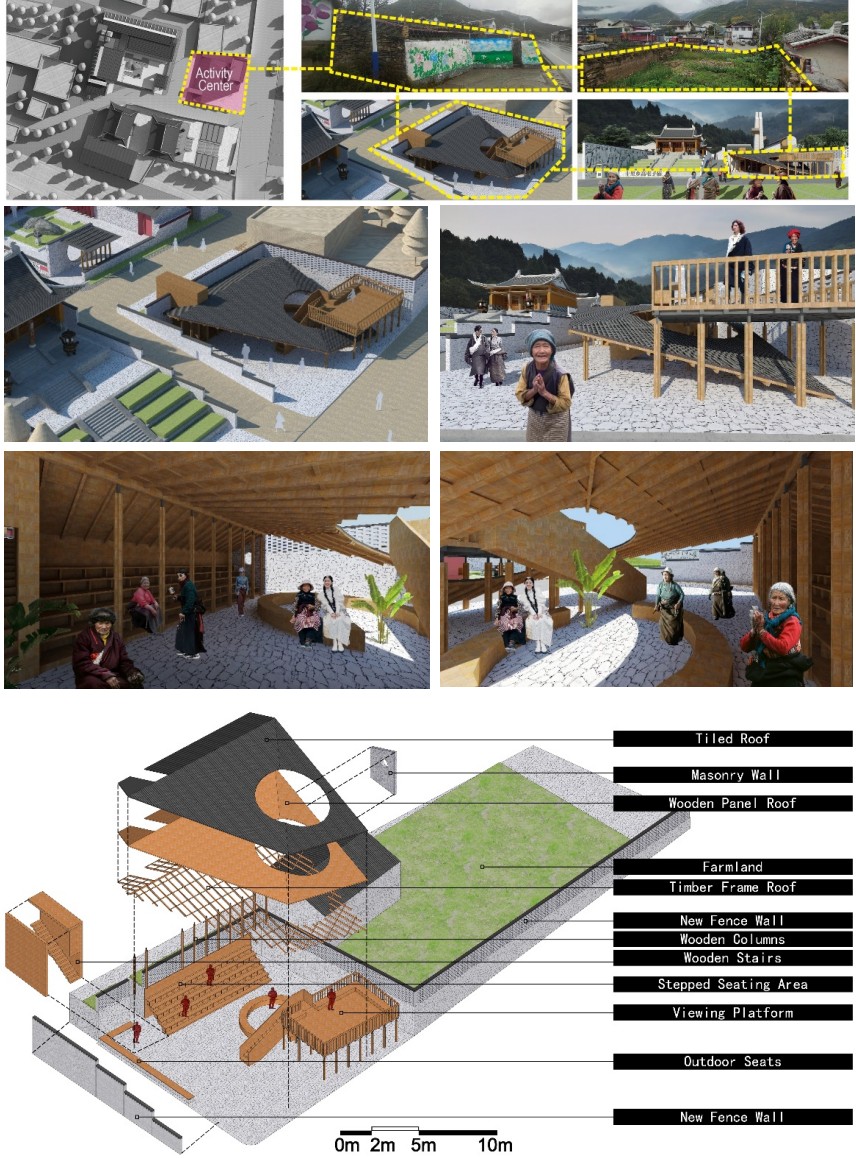

**Figure 7.** Public space reactivation with Villager Activity Center design details (rendering by the authors).

Compared to the previously described simple shed of 4 square meters, the design proposal increases the area of public activity space to 160 square meters, significantly improving the area in size and function. Across the national highway, a centralized parking lot is newly constructed, adding 15 parking spaces (including 2 bus parking spaces and 13 regular parking spaces), which can help alleviate the issue of vehicles being parked randomly in the village and encroaching on public spaces.

## 4. Discussion: Traditional–Modern Culture Integration

Against the backdrop of rapid modernization and cultural preservation, rural revitalization practices in western China have become increasingly common. A great many villages are mechanically transformed with modern architectural styles, materials, and techniques, which results in "a thousand villages looking exactly the same". However, restoring traditional villages into their original appearances is not only impossible but also unrealistic. Cultural preservation should not dwell on copying or restoring history in its original form but rather on engaging in a dialogue with history.

Traditional Jiarong Tibetan dwellings are typically made of stone with the second and third floors constructed using wooden joints. The design of the exhibition space for Xue Tao's poetry and Jiarong Tibetan culture aims to explore a way to integrate Jiarong Tibetan culture into modern life. The design proposes using concrete with rough stone cladding for the exterior walls, which can be left in its natural yellowish color or painted with white coating. The roof will be constructed with dark gray tiles. The ground floor of the building will be formed by an earth-mound wall foundation, creating interior spaces, while the exterior walls will have ventilation ducts and small windows. Traditional wooden wall structures are recommended for use in the second floor of the building. This type of construction provides good insulation and thermal insulation, suitable for the climate characteristics of the plateau region. The doors and windows are suggested to be primarily made of wood, with panel doors that open horizontally, which is conducive to future commercial function transformations.

As Figure 8 indicates, the doors and windows will be embedded in the walls, adding variations in light and shadow to the facade. The second-floor terrace railing can be made of woven materials, creating a lightweight and transparent effect. The courtyard walls and partition walls will be a combination of solid walls, perforated brick walls, and movable door panels, blending tradition with modernity. Traditional painted or carved decorations can be applied to the doors and windows, and colorless glass will be used, avoiding reflective or colored glass. The building's color scheme will use traditional rough stones, tiles, and natural wood colors, or be painted in white, deep red, yellow, etc.

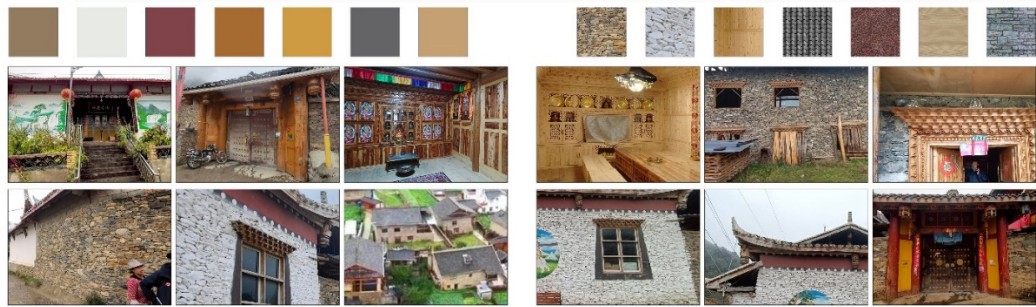

**Figure 8.** Traditional architectural colors and building materials in Gaotunzi Village.

As Figure 9 illustrates, the structural system of the Xue Tao Cultural Museum is proposed to adopt a heavy steel and light steel composite structure: a multistory steel frame structure with heavy steel providing load-bearing support and light steel serving as the enclosure system. Some building components of the museum can be constructed using a semiprefabricated construction method, involving prefabrication and onsite assembly. The exterior walls of the museum will combine traditional Tibetan rough stone

masonry techniques with modern frame structures, with rough stone walls built outside the supporting structure of the brick walls. Onsite assembly of building components can effectively increase the installation speed of the main construction, shorten the construction period, reduce the weight of the structure, decrease material waste, and minimize the impact and damage of construction on the natural environment, which is beneficial for environmental conservation.

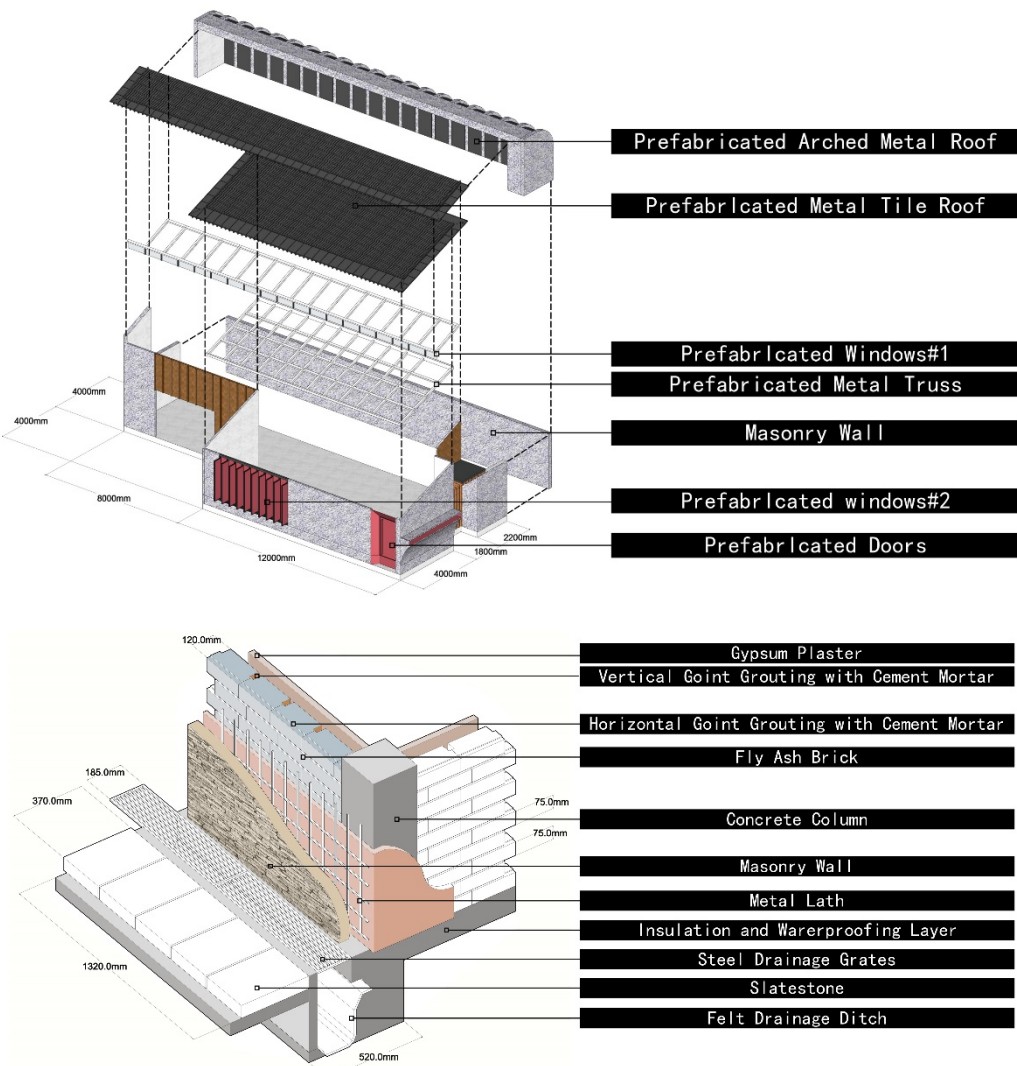

**Figure 9.** Schematic configuration of semiassembled structure and semiprefabricated buildings.

Various elements are combined in the core space of traditional Tibetan settlements in Songpan. Religious facilities such as temples, lama towers, and prayer rooms, among other material settlement elements, are constructed in the most advantageous locations within the settlement space, generally not adjacent to water. The spatial positioning of other material settlement elements with different attributes is relatively flexible. Settlement areas often have clear boundaries or restrictions. The natural landscape forms the outer foundational environment of the settlement, while cultivated land, grasslands, mountains, water bodies, and buildings are the basic elements that outline the boundaries of the settlement.

The architectural layout of traditional villages typically revolves around a central square, with houses arranged in concentric circles around it. The central square serves as a place for community activities and gatherings, symbolizing the cohesion of the community. The houses are arranged based on family and kinship relationships, creating a closely connected community. The material elements of the settlement serve different functions

based on the activities of the residents. Elements such as parking lots, recreational spaces, lama towers, prayer rooms, temples, and other public spaces are clustered around the central buildings, forming a surrounding layout. These spaces are also important places for communication and interaction among community residents.

Residential buildings are the fundamental entities that make up the settlement and are not combined with other functional elements due to requirements for privacy and security. The layout of Jiarong Tibetan residential buildings typically revolves around a central courtyard, with various functional areas surrounding it. The courtyard serves as a place for family activities and social interactions, symbolizing a microcosm of the city. The core space of Gaotunzi Village, formed by temples, watchtowers, cultural centers, and community activity centers, engages in a dialogue between modern design techniques and construction technologies with traditional and historical elements. By respecting the spatial layout of Songpan traditional settlements and the architectural wisdom of Jiarong Tibetan buildings, it can provide us with ideas for the protection and renovation design practices of other traditional Tibetan villages in the western Sichuan plateau.

## 5. Conclusions

The paradoxical problem between fast modernization and cultural preservation arouses challenges on new insights into green construction and sustainable development strategies in China. How to strike a balanced cultural–modern rural revitalization has become a research priority, especially for cultural and historical villages. In this paper, taking Gaotunzi Village, a traditional ethnic village in western high-altitude plateau as an illustrative example, the typical green design manner and optimization strategy are proposed for cultural and architectural heritage preservation. The detailed architecture and structure design is conducted for both old temple retrofitting and new construction, with consideration of retaining traditional building colors, styles, and materials. Based on the typical case, the main findings and enlightenments follow:

(1) Key design strategy of Gaotunzi Temple lies in the protection and exhibition of the historical buildings (i.e., main hall). The retrofitting manner combines the religious beliefs and local living customs, fully preserving the architectural heritage itself. Apart from basic renovation and maintenance, several poorly constructed structure and auxiliary elements are partially demolished and refurnished for restoring the traditional ceremonial spatial order and improving the functions of the temple.

(2) With the space concentrated on the temple, the expansion design involves new construction, including Xue Tao Cultural Museum, Village History Museum, public toilets, and a community activity center. Such land planning not only effectively makes full use of abandoned or idle buildings, but also substantially enriches public spaces with leisure and communication services for local villagers.

(3) Integrated traditional–modern design style respects the spatial layout of the Songpan regional and Jiarong Tibetan architectural wisdom, incorporating forms such as watchtowers, courtyard houses, sloping roofs, timber structures, and stone masonry to modernize the cultural elements of the architectural style, form, color, and scale of the Qiang and Tibetan settlements in western Sichuan's Aba region. This retrofitting program significantly extends the skyline and landscape, with emphasis on symbolic and iconic qualities, facilitating potential transformation and further development of Gaotunzi Village towards ecotourism and cultural hospitality.

(4) Application of prefabricated construction techniques, including steel structures, premanufactured doors, windows, roof panels, etc., can improve construction efficiency and quality, with possibly little impact on historic heritage parts. It provides a good reference and retrofitting prototype for sustainable development in traditional Tibetan ethnic villages in western Sichuan's Aba region.

The present work only demonstrates a typical case to illustrate the green retrofitting manners and strategies for architectural heritage preservation and traditional village renovation. The specific design details presented may not be applicable for other situations

because of the massive diversity of ethnic groups, architectural forms, building materials, cultural styles, and living standards. These limitations also arouse in-depth investigations for future study; however, the planning idea and design approaches used here with traditional–modern integration considerations are general, which are regional adaptively applicable for similar sites. This work can provide a typical design reference and application prototype for rural construction and modernization with local heritage preservation considerations, especially for traditional villages in developing countries.

**Author Contributions:** Conceptualization, K.X. and Y.Z.; methodology, K.X.; software, K.X. and W.H.; validation, W.H. and Y.Z.; formal analysis, Y.Z.; investigation, K.X.; resources, W.H.; data curation, Y.Z.; writing—original draft preparation, K.X.; writing—review and editing, W.H. and Y.Z.; visualization, K.X.; supervision, Y.Z. All authors have read and agreed to the published version of the manuscript.

**Funding:** This research is funded by the Sichuan Science and Technology Research Program (No. 2022NSFSC1081), Central University Basic Research Business Fee Special Fund Project (No. 2022SQN05), and Introduction of Talent Research Initiation Fund Funded Project (No. RQD2022030).

**Institutional Review Board Statement:** Not applicable.

**Informed Consent Statement:** Not applicable.

**Data Availability Statement:** Data, original images, and detailed design model materials are available on request.

**Conflicts of Interest:** The authors declare no conflict of interest. The funders had no role in the design of the study; in the collection, analyses, or interpretation of data; in the writing of the manuscript; or in the decision to publish the results.

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
