# Peer review of "Architectural Heritage Preservation for Rural Revitalization: Typical Case of Traditional Village Retrofitting in China"

_sustainability, doi:10.3390/su16020681_

Round 1

Reviewer 1 Report

Comments and Suggestions for Authors

see the file uploaded

Comments on the Quality of English Language

English should be revised. Some of the word chosen is not always appropriate and the meaning of some sentences is not always very clear.

Author Response

The authors sincerely appreciate all the rewarding suggestions on our paper improvement. We have made significant revisions accordingly. Please see the revised manuscript and the attachment for point-by-point response.

Reviewer 2 Report

Comments and Suggestions for Authors

After congratulating for the development of the work, it is suggested:

1. Review the structure of the abstract, the methodology, results and conclusions are not presented in a clear way. The text is entirely introductory. The case study is not mentioned either.

2. Fundamental documents such as the Charter of Built Vernacular Heritage have not been considered for the study. It is recommended to resort to this type of worldwide documents.

3. It is recommended to diversify geographically the sources of consultation. The West is practically excluded.

4. It is recommended to include sources of applicable technical standards.

5. Segments 2.1, 2.2, 2.3, 2.4 are descriptive contents of the case study and do not reflect the methodology applied in the study presented.

6. The quality of the texts in the figures should be revised, they are difficult to read. In addition, the parts in a figure with more than 1 element should be clearly recognized, as in the case of figure 5, 6 and others.

7. Figure 9, among other things, should include dimensioning information.

8. Technical figures should include scales.

9. The article's approach to ecology should be reviewed and be objective. There is a lot of rhetorical content and the objective evidence on the subject is not clear.

Comments on the Quality of English Language

Not applicable

Author Response

(The authors gave the same response as above.)

Reviewer 3 Report

Comments and Suggestions for Authors

This is an important topic worldwide and it will continue to be important in the years to come. The authors' aims -- to describe ideas for retrofitting a traditional Chinese village -- are fine. 

However, the set-up of the paper and description of the work are unclear and the narrative does not follow a clear and logical course. This appears to be an architect's consultant report to the village, but that is not stated in the beginning. The authors should clearly explain what this is and why it is important. 

The introductory narrative uses very little literature; it would help the piece if the authors could find literature that refers to precisely the kind of preservation work as the consultant project they wish to describe. In other words, state the key questions in a broader discussion at the outset, and then present this case as a response to the issues raised in the literature. 

Also, the labels on the images are illegible. The images could help to make the case, but all of them need to be simpler and easier to read. 

The authors need to define what they mean by the word "green," which is used repeatedly in the piece. 

Comments on the Quality of English Language

The manuscript is difficult to read. It needs a complete edit by someone who has native-level competence in English. This will possibly identify some places where the ideas themselves are unclear, and that will help the piece. 

Author Response

(The authors gave the same response as above.)

Reviewer 4 Report

Comments and Suggestions for Authors

Dear authors thanks so much for you contribution. The article is very interesting, although I would like to suggest some directions for its improvement.

Given the complexity of the topic and its various approaches to the posed problem, it would be necessary to design a flowchart that clarifies the entire research process developed in this research article.

Can you explain with greater precision what the concept of acupuncture approach is in this case?

Figure 8 is very interesting as a model for approaching the conclusions of the work. However, the color models, their tones, and material textures are not identified in relation to the images representing architectural spaces and/or surfaces. Could you include some graphics (e.g., arrows, lines, letters) that appropriately relate the colors and textures?

The bibliographic references are quantitatively insufficient. Is need to be implemented and expanded.

Author Response

(The authors gave the same response as above.)

Round 2

Reviewer 1 Report

Comments and Suggestions for Authors

Comments on the Quality of English Language

Minor editing is required

Author Response

Dear Reviewer,

The authors sincerely appreciate the positive comments on our last revision. We have spared no time and efforts to make further improvements. We have carefully addressed each suggestion this time. Please see the attached point-by-point responses and revised manuscript (v2).

Merry Christmas in advance

Reviewer 2 Report

Comments and Suggestions for Authors

Hello, ajsutes are appreciated, however, it is commented that:

1. The abstract should explicitly state the name of the case studies. This also applies to the methodology followed and the clear consequences (conclusions) of its application. Avoid rhetorical content.

2. Review the length of the paragraphs, lengths longer than 12 lines are difficult to understand.

3. The methodology has not considered the inclusion of research techniques and instruments, i.e., how the research is actually done and what technical resources it uses. Avoid rhetorical content.

4. Much of the discussion content is actually a description of the objects of study; the findings are not questioned or defended.

5. References are kept mainly in and around China.

Comments on the Quality of English Language

Review the length of sentences and paragraphs, as well as the use of punctuation marks.

Author Response

(The authors gave the same response as above.)

Reviewer 3 Report

Comments and Suggestions for Authors

The manuscript is much improved and is much more interesting to read now. Thank you. 

Comments on the Quality of English Language

The writing in the revised version, specifically the structure of sentences and grammar in general, is much better than in the original. Errors in word usage remain a minor problem. 

Author Response

Dear Reviewer,

The authors sincerely appreciate the positive comments on our last revision. We have spared no time and efforts to make further improvements, according to the other reviewers' suggestions. Please see the revised manuscript (v2).

Merry Christmas in advance

Round 3

Reviewer 1 Report

Comments and Suggestions for Authors

The paper has been revised according to the suggestions given.

Only few minor text revisions are need.

Comments on the Quality of English Language

Minor editing of English language are required